# Unmet Medical Needs and Future Perspectives for Leiomyosarcoma Patients—A Position Paper from the National LeioMyoSarcoma Foundation (NLMSF) and Sarcoma Patients EuroNet (SPAEN)

**DOI:** 10.3390/cancers13040886

**Published:** 2021-02-20

**Authors:** Bernd Kasper, Annie Achee, Kathrin Schuster, Roger Wilson, Gerard van Oortmerssen, Rebecca A. Gladdy, Matthew L. Hemming, Paul Huang, Matthew Ingham, Robin L. Jones, Seth M. Pollack, Denise Reinke, Roberta Sanfilippo, Scott M. Schuetze, Neeta Somaiah, Brian A. Van Tine, Breelyn Wilky, Scott Okuno, Jonathan Trent

**Affiliations:** 1Mannheim University Medical Center, University of Heidelberg, 68167 Mannheim, Germany; 2National LeioMyoSarcoma Foundation (NLMSF), Denver, CO 80222, USA; annieachee@aol.com; 3Sarcoma Patients EuroNet, SPAEN, 61200 Wölfersheim, Germany; Kathrin.schuster@sarcoma-patients.eu (K.S.); rogleswil@me.com (R.W.); gerard.vanoortmerssen@gmail.com (G.v.O.); 4Department of Surgery, Mount Sinai Hospital, Toronto, ON M5G 1XS, Canada; Rebecca.Gladdy@sinaihealth.ca; 5Dana-Farber Cancer Institute, Boston, MA 02215, USA; mhemming@partners.org; 6Institute of Cancer Research, London SM2 5NG, UK; Paul.Huang@icr.ac.uk (P.H.); robin.jones@rmh.nhs.uk (R.L.J.); 7Department of Medicine, Columbia University School of Medicine, New York, NY 10032, USA; mi2337@cumc.columbia.edu; 8Royal Marsden Hospital, London SW3 6JJ, UK; 9Northwestern Medicine, Feinberg School of Medicine, Chicago, IL 60611, USA; seth.pollack@northwestern.edu; 10Sarcoma Alliance for Research through Collaboration (SARC), Ann Arbor, MI 48105, USA; dreinke@sarctrials.org; 11Fondazione IRCCS Istituto Nazionale dei Tumori, 20133 Milan, Italy; Roberta.Sanfilippo@istitutotumori.mi.it; 12Michigan Medicine Sarcoma Clinic, Rogel Cancer Center, Ann Arbor, MI 48109, USA; scotschu@med.umich.edu; 13Department of Sarcoma Medical Oncology, The University of Texas MD Anderson Cancer Care Center, Houston, TX 77030, USA; NSomaiah@mdanderson.org; 14Barnes and Jewish Hospital, Washington University in St. Louis, St. Louis, MO 63110, USA; bvantine@wustl.edu; 15Department of Sarcoma Medical Oncology, Anschutz Medical Campus, University of Colorado, Aurora, CO 80045, USA; breelyn.wilky@ucdenver.edu; 16Division of Medical Oncology, Mayo Clinic, Rochester, MN 55905, USA; okuno.scott@mayo.edu; 17Sylvester Comprehensive Cancer Center, University of Miami, Miami, FL 33136, USA; JTrent@med.miami.edu

**Keywords:** leiomyosarcoma, NLMSF, SPAEN, treatment, research

## Abstract

**Simple Summary:**

In this position paper, we aim to summarize state-of-the-art treatments for patients with leiomyosarcomas in order to identify knowledge gaps and current unmet needs, thereby guiding the community to design innovative clinical trials and basic research and close these research gaps. This white paper arose from a leiomyosarcoma research meeting in October 2020 hosted by the National LeioMyoSarcoma Foundation (NLMSF) and Sarcoma Patients EuroNet (SPAEN).

**Abstract:**

As leiomyosarcoma patients are challenged by the development of metastatic disease, effective systemic therapies are the cornerstone of outcome. However, the overall activity of the currently available conventional systemic treatments and the prognosis of patients with advanced or metastatic disease are still poor, making the treatment of this patient group challenging. Therefore, in a joint effort together with patient networks and organizations, namely Sarcoma Patients EuroNet (SPAEN), the international network of sarcoma patients organizations, and the National LeioMyoSarcoma Foundation (NLMSF) in the United States, we aim to summarize state-of-the-art treatments for leiomyosarcoma patients in order to identify knowledge gaps and current unmet needs, thereby guiding the community to design innovative clinical trials and basic research and close these research gaps. This position paper arose from a leiomyosarcoma research meeting in October 2020 hosted by the NLMSF and SPAEN.

## 1. Introduction

Soft-tissue sarcomas (STS) represent a highly heterogeneous group of mesenchymal malignancies comprising approximately 175 distinct histological subtypes. Within these, leiomyosarcoma (LMS) is one of the most frequent subtypes, accounting for approximately 10–20% of all STS. LMS typically occurs in middle-aged or older adults, with a female predominance. Younger patients may also be affected, especially in the context of LMS predisposition genetic syndromes such as Li-Fraumeni or Hereditary Leiomyomatosis and Renal Cell Cancer syndromes. LMS is thought to form from the smooth muscle or their precursor cells, and thus can arise anywhere in the body with a predilection for tumors in the uterus, the retroperitoneum and the extremities [1]. Anatomically, LMS can be divided into “extra-uterine” (retroperitoneal, inferior vena cava or renal vein, gastrointestinal, extremity, or subcutaneous) and “uterine” LMS, each with distinct clinicopathological characteristics [2]. Overall, with complete resection of primary LMS, the main pattern of failure is distant metastasis as recent histotype-specific series from expert centers report lower local recurrence rates compared to other common STS such as liposarcoma [3]. Retrospective data suggest that there may be differential sensitivity to chemotherapy, but this still requires confirmation [4,5].

## 2. Pathological and Clinical Features

LMS are smooth muscle malignant mesenchymal tumors [6]. More differentiated LMS show spindle cell morphology, with sharply demarcated vertical crossing tumor bundles analogous to normal smooth muscle; however, morphologic variants are common, most frequently an epithelioid, myxoid, or pleomorphic subtype [6]. Immunohistochemistry shows tumor cells expressing smooth muscle-specific antigens, pan-muscle actins, and desmin or h-caldesmon in over 70% of LMS. Grading of LMS is usually performed according to the 3 tier Fédération Nationale des Centres de Lutte Contre le Cancer (FNCLCC) system, which does not apply to uterine LMS.

Clinical symptoms are mostly non-specific and mainly caused by the tumor location. Generalized symptoms such as fever, fatigue, weight loss, or gastrointestinal symptoms are rare and usually associated with advanced disease. Patients with retroperitoneal LMS are often asymptomatic and many are incidentally discovered. In advanced cases, patients may notice an enlarged abdomen. Uterine LMS may be associated with acyclic bleeding. Diagnosis and staging of patients with LMS are in line with the general recommendations for STS and visceral sarcomas [7]. The overall management of patients with LMS including confirmation of the diagnosis by expert pathologic review and consideration of surgical resection of primary disease should be part of a multidisciplinary team in a sarcoma reference center.

## 3. Localized Soft-Tissue (Non-Uterine) Leiomyosarcomas

Surgery remains the cornerstone in the management of patients with localized LMS and the standard surgical procedure is a wide excision with negative margins (R0) [7]. Specialized considerations include the need for vascular reconstruction in LMS that arise from central veins such as the inferior vena cava. In case of R1 (microscopic positive margins) or R2 (macroscopic positive margins) positive resections, re-operation in experienced hands may be considered, possibly following preoperative treatments if adequate margins cannot be achieved, or surgery would be associated with unacceptable morbidity. The 5 year local and distant recurrence rates for primary LMS are 10–20% and 30–40% for high-grade tumors, respectively. Significant independent predictors for local recurrence are size and margin, whereas predictors for distant recurrence are size and grade [8]. Improved local control rates are most likely multifactorial: improved preoperative diagnostic care using MRI for pelvic and extremity tumors and cross-sectional CT scans for intra-abdominal LMS as well as routine use of core biopsies to diagnose the disease. Planned surgical resections by experts in STS are a cornerstone of local control by obtaining margin negative resections. Additionally, borderline resectable tumors would be considered for neoadjuvant treatment including chemotherapy and/or radiotherapy. Further, in most centers, advanced radiation techniques are employed such as IMRT. Why exactly LMS have a high rate of metastatic failure, is an active area of translational research.

In patients with high-risk extremity LMS (G2–G3, deep > 5 cm lesions), adjuvant or neoadjuvant radiation therapy can be administered in addition to surgery. In patients with LMS that is retroperitoneal or in the pelvis, especially if borderline resectable, consideration should be given to neoadjuvant radiation. We have learned from the European Organization for Research and Treatment of Cancer (EORTC) STRASS trial that neoadjuvant radiotherapy for LMS is not likely beneficial for resectable tumors; if borderline resectable, it can be considered [9]. In selected cases, radiation therapy may be added to the management of patients with low- or high-grade superficial, ≥5 cm and low-grade, deep, <5 cm STS after multidisciplinary discussion. R1 and R2 resections are followed by radiation therapy if margins cannot be rescued by re-resection [7].

Adjuvant chemotherapy is not globally accepted as the standard treatment strategy for adult patients with LMS as an extrapolation from studies involving all STS. Although two meta-analyses and individual prospective, randomized trials support the use of adjuvant chemotherapy, other randomized trials found no benefit when chemotherapy was added to surgery with or without radiation. Thus, prospective trial results are conflicting and currently universal consensus does not exist [7]. However, adjuvant chemotherapy may be proposed as an option for patients with LMS at high risk for local or distant recurrence. As with any treatment for sarcomas, decisions should always be made in consultation with the patient and treatment in the context of a clinical trial when possible and appropriate.

Neoadjuvant chemotherapy has the same potential benefits as adjuvant chemotherapy. Additionally, neoadjuvant chemotherapy may facilitate limb- or organ-sparing surgery, initiate treatment of micrometastatic disease earlier and avoid very morbid surgery for the rare patient who develops metastatic, chemoresistant disease on therapy. Similar to adjuvant chemotherapy, there is no consensus regarding the role of neoadjuvant chemotherapy. Neoadjuvant chemotherapy as well as radiation therapy may be considered for patients with high-risk extremity/trunk LMS (lesion diameter >5 cm, tumor deep to fascia, adjacent to bone or neurovascular structures, invasion of skin). In a population of patients with high-risk extremity/trunk STS including LMS, no benefit of neoadjuvant histotype-tailored chemotherapy regimens over standard chemotherapy could be demonstrated. Hence, when neoadjuvant chemotherapy is to be administered, the combination of doxorubicin plus ifosfamide remains the regimen of choice [10].

The efficacy of neoadjuvant chemotherapy in retroperitoneal LMS (and liposarcomas) is currently being evaluated in the EORTC/Soft Tissue and Bone Sarcoma Group (STBSG) STRASS-2 trial (NCT04031677) in patients with resectable retroperitoneal sarcomas.

Hence, neoadjuvant therapy is an option for the patient if the LMS is at risk for local or distant recurrence, borderline resectable, or function preservation is an important goal [7].

## 4. Localized Uterine Leiomyosarcomas

For patients with uterine LMS, en bloc total hysterectomy is the standard surgical approach and should be carried out by a (gynecologic) surgeon specifically trained in the treatment of this disease. Of note, almost all uterine LMS are diagnosed after surgery for supposed leiomyoma as there are no preoperative radiological findings that reliably differentiate ordinary leiomyomas from leiomyoma variants and from uterine LMS. Laparoscopy/assisted or robotic surgery are feasible as long as the tumor is resected with the same criteria as for open surgery [7]. Fertility-preserving surgery is not recommended in women at reproductive age. Particularly, in premenopausal women, the value of bilateral salpingo-oophorectomy is not established. Lymph node involvement occurs in less than 3% of all cases and lymphadenectomy has not been demonstrated to be useful in the lack of macroscopic involvement [7]. The use of preoperative diagnostic biopsy is rarely utilized although could impact treatment recommendations in the setting of low-grade or non-malignant tumors such as leiomyoma, inflammatory myofibroblastic tumor, endometrial stromal sarcoma and small grade 1 LMS.

A prospective randomized controlled trial did not show any benefit from radiotherapy for uterine LMS [11].

The role of adjuvant chemotherapy in uterine LMS is still uncertain [7]. Uncontrolled prospective trials suggested a benefit of adjuvant chemotherapy in comparison with external controls for gemcitabine plus docetaxel followed with or without doxorubicin [12,13]. A prospective randomized trial investigating adjuvant gemcitabine plus docetaxel followed by doxorubicin monotherapy versus no chemotherapy was stopped due to poor recruitment. Adjuvant treatment should be considered in case of tumor rupture during surgery in view of the associated poor prognosis [14].

## 5. Advanced/Metastatic Soft-Tissue (Non-Uterine) Leiomyosarcomas

Treatment of advanced and/or metastatic LMS is a major challenge and, for most patients, maintaining quality of life and improving outcome are the main goals. In order to select the right treatment option for each patient, definition of treatment goals and individual expectations is essential. Patient age, performance status, comorbidities, disease stage, and tumor volume have to be taken into account to define the best treatment strategy in terms of symptom alleviation in highly symptomatic patients or life prolongation in asymptomatic individuals.

Standard first-line chemotherapy for STS consists of anthracycline-based regimens, and doxorubicin is the first-line chemotherapy of choice in patients with advanced LMS [7]. In a randomized phase 3 trial, doxorubicin plus ifosfamide showed a significantly higher response rate and longer progression-free survival (PFS) compared to single-agent doxorubicin, but no significant difference in overall survival (OS) [15]. In patients with LMS, the combination of doxorubicin plus dacarbazine is an option for multiagent first-line chemotherapy. In a retrospective analysis median PFS and OS were 15.1 and 33.9 months, respectively. Partial response was achieved in almost one-third (27%) of patients, and the clinical benefit rate was 95% [16]. Although ifosfamide is an effective therapy for women with uterine LMS, it appears to be less effective for patients with extra-uterine LMS [17]. An exploratory retrospective EORTC/STBSG analysis of a subset of patients diagnosed with LMS showed no benefit from ifosfamide, with a significantly decreased OS compared to doxorubicin monotherapy [18]. Promising data have been reported for the first-line combination of doxorubicin plus trabectedin in extra-uterine and uterine LMS (LMS-02) [19]; however, final results from the randomized phase 3 trial comparing this combination versus doxorubicin alone (LMS-04) are awaited.

Trabectedin is registered for the treatment of advanced STS (including LMS) after failure of doxorubicin with or without ifosfamide in second line or later, or for patients “unsuited” to receive these agents. Chemosensitivity to trabectedin has been noted in different STS subtypes, but best responses have been observed in LMS, liposarcomas and in translocation-associated sarcomas. Based on the results of a randomized, multicenter, phase 2 study comparing two dose schedules, trabectedin monotherapy was approved by the European Medicines Agency. A significant benefit was demonstrated for the 24 h infusion every third week. Two-thirds of all patients treated in this trial were diagnosed with LMS and the median OS was 13.9 months with manageable side effects and no cumulative toxicity [20]. Trabectedin was approved by the United States Food and Drug Administration based on the results of a trial randomizing patients with pretreated, advanced LMS and liposarcoma, to receive trabectedin or dacarbazine. The trial demonstrated significantly longer PFS in the trabectedin arm compared to the dacarbazine arm, but no significant difference in OS [21]. The clinical benefit with trabectedin, especially for patients with LMS, was supported by the results of a worldwide expanded-access program with trabectedin in patients with advanced STS following failure of prior chemotherapy. A median OS of 16.2 months was observed in 318 patients with LMS [22].

In view of the section below on “LMS-specific studies”, it should be highlighted that the phase 3 eribulin trial included LMS and liposarcoma patients. Interestingly, higher response rates and rates of disease control were seen with dacarbazine for the LMS cohort in comparison to liposarcoma patients; this may have been the reason that eribulin was deemed ineffective for the LMS population [23]. Hence, dacarbazine is a reasonable choice to consider in the refractory setting for LMS, also in combination with gemcitabine being well tolerated and given on a convenient schedule [24]. There is considerable clinical and biological heterogeneity within the LMS histological subtype. Therefore, heterogeneity is a major challenge even for subtype-specific trials. The primary analysis for the phase 3 ANNOUNCE trial evaluating olaratumab in combination with doxorubicin was for “all STS” but also for “LMS” [25].

Two randomized studies comparing the efficacy of gemcitabine plus docetaxel versus gemcitabine monotherapy reported divergent findings in patients with relapsed or metastatic LMS [26,27]. In a subsequent pooled analysis, no significant improvement of response rate and PFS could be demonstrated by the addition of docetaxel for LMS [28]. In a randomized phase 3 trial in first-line advanced STS, no significant difference in response rate, PFS and OS was observed between single-agent doxorubicin and gemcitabine plus docetaxel, although doxorubicin was better tolerated. A planned analysis for the LMS cohort showed similar findings [29].

Other potential strategies including conventional chemotherapeutic agents comprise gemcitabine re-challenge in LMS, and other combination schedules such as gemcitabine plus dacarbazine [24] or gemcitabine plus vinorelbine [30].

Pazopanib has been registered in Europe and the United States for selected subtypes of advanced, adult STS patients including LMS after prior chemotherapy for advanced and/or metastatic disease. The PALETTE study included 165 patients with LMS. Pazopanib was shown to significantly prolong PFS. However, this did not translate into a statistical difference in OS for the subgroup of patients with LMS compared to placebo [31].

For oligometastatic disease, other modalities such as radiotherapy or surgery can be applied. Other local therapies such as radiofrequency ablation may also be considered for liver metastases.

## 6. Advanced/Metastatic Uterine Leiomyosarcomas

In patients with indolent, low-volume metastatic disease, endocrine therapy can be considered. A single-arm phase 2 trial in patients with uterine LMS expressing estrogen receptor (ER) and/or progesterone receptor (PgR) evaluated letrozole and met predefined criteria for activity in uterine LMS. Patients with longest PFS rate were those whose tumours strongly and diffusely expressed ER and PgR [32]. More recently, there are reports of activity of endocrine therapy in patients with low-grade uterine LMS [33].

Doxorubicin alone or in combination with ifosfamide remains the gold-standard first-line treatment for STS and, therefore, also for uterine LMS (see above for details on doxorubicin combinations).

In a second-line or greater, single-arm study, the combination of gemcitabine plus docetaxel demonstrated an overall response rate of 27%, median PFS of 6.7 months, and median OS of 14.7 months in 51 patients with advanced uterine LMS after prior cytostatic chemotherapy. However, this combination was associated with hematological toxicity and other specific toxicities including neurotoxicity induced by docetaxel and pulmonary toxicity caused by gemcitabine [34]. Hensley et al. further investigated the role of gemcitabine plus docetaxel as first-line treatment in 42 women with advanced uterine LMS. In this phase 2 single-arm study objective responses were observed in 36% of patients. Median PFS was 4.4 months and median OS was 16.1 months [35]. In a randomized study, both single-agent gemcitabine and gemcitabine plus docetaxel were found to be effective second-line therapies for LMS, with a 3-month PFS rate of 40% for both uterine and extra-uterine LMS (TAXOGEM). Single-agent gemcitabine yielded results similar to those of gemcitabine plus docetaxel in this trial, but patients using single-agent gemcitabine experienced less toxicity [27].

Trabectedin is an additional valid treatment strategy. In a retrospective analysis of 66 patients who progressed on a median of three previous cytotoxic lines including anthracycline-based chemotherapy ± ifosfamide and gemcitabine ± docetaxel, 16% achieved partial response, and 35% stable disease, for a disease control rate of 51%. Median PFS was 3.3 months and median OS was 14.4 months. Trabectedin was associated with good tolerability and lack of cumulative toxicity [36]. In a post hoc subset analysis of patients with uterine LMS who had received prior anthracycline therapy, trabectedin treatment resulted in significantly longer PFS versus dacarbazine, with an acceptable safety profile. There was no difference in OS [37].

In a retrospective analysis of two EORTC/STBSG trials, the outcome of advanced and/or metastatic patients with uterine LMS treated with pazopanib was investigated. Out of 44 patients, 68% achieved clinical benefit (five partial responses, 25 stable diseases). Median PFS was 2.9 months versus 4.5 months for other STS subtypes; median OS was 17.5 versus 11.1 months. Pazopanib showed similar efficacy in uterine as in extra-uterine LMS [38]. Table 1 illustrates an overview of key studies on the current clinical management of advanced/metastatic patients with LMS.

## 7. Unmet Medical Needs, Therapeutic Gaps and Future Perspectives

As illustrated above, the guidelines for adjuvant/neoadjuvant therapy in localized LMS warrant further clinical development and the overall effectiveness of the currently available systemic treatment options for extra-uterine and uterine patients with LMS in the advanced and/or metastatic setting is limited, thus, patients’ overall prognosis remains poor. Therefore, in a joint effort together with patient networks and organizations, namely SPAEN, the international network of sarcoma patients organizations, and the NLMSF in the United States, we aim to identify unmet medical needs and gaps in the present treatment standards and overall disease management driving the community to design innovative clinical trials and basic research projects in order to close these gaps.

In 2019, SPAEN has started a project to form a “Patient Powered Research Network” in order to help develop a clearer strategy and direction to ultimately improve outcomes for sarcoma patients. An international questionnaire was distributed among sarcoma patients with the aim to capture the patients’ view on the sarcoma research agenda covering the categories diagnosis, treatment, support, quality of life, survivorship and end of life. In total, 265 patients responded. A subset of responses of 25 patients with LMS from eight countries (UK, USA, Canada, Australia, Germany, France, Spain and The Netherlands) has been analyzed to gain insight into the specific problems and needs of patients with LMS complementing our consensus process of clinicians and researchers for the following identified unmet medical needs:Design LMS-specific studies for evaluating sequence and combinations of available systemic therapies: Evidence-based data for LMS mainly comes from clinical trials open for the recruitment of a variety of heterogeneous STS subtypes; there are hardly any prospective trials exclusively designed for patients with LMS. As an example, the North Eastern German Society of Gynaecological Oncology is currently evaluating the role of pazopanib versus pazopanib plus gemcitabine in the treatment of advanced or metastatic uterine LMS including carcinosarcomas in an ongoing prospective randomized controlled phase 2 trial (PazoDoble; NCT02203760). The French Sarcoma Group has conducted a randomized phase 3 study comparing the efficacy of doxorubicin plus trabectedin followed by trabectedin versus doxorubicin alone in patients with LMS from which the final results are eagerly awaited (LMS-04; NCT02997358). The EORTC/STBSG is currently setting up an open-label, randomized, phase 2 study on doxorubicin, doxorubicin plus dacarbazine, or gemcitabine plus dacarbazine for first-line treatment of advanced patients with LMS (DODECANESO) based on a published retrospective STBSG analysis [41]. There is a clear need for large, international, randomized and single-arm LMS histology specific clinical trials, with an underlying biological rationale.Explore new therapeutic avenues: In addition to the evaluation of the activity of conventional chemotherapeutic agents for patients with LMS, new treatment avenues need to be explored. Currently, there are a number of ongoing trials exploring the possible value of immunotherapeutic agents in patients with LMS (see Chapter 4). Anlotinib is being evaluated in a randomized phase 3 trial with a distinct LMS cohort (APROMISS; NCT03016819). Another tyrosine kinase inhibitor (TKI), surufatinib, is also being tested in LMS. Another strategy aims to evaluate the BMI1 inhibitor PTC596 in combination with dacarbazine in participants with advanced LMS from PTC Therapeutics (NCT03761095). Based on recent basic research results for LMS [42,43], a number of trials are currently evaluating PARP inhibition combined with chemotherapy. One trial is evaluating olaparib plus trabectedin versus doctor’s choice in various solid tumors harboring deficiency in DNA repair but is not specific to sarcoma (NCT03127215). A phase 1B trial of the combination of olaparib and trabectedin in patients with previously treated advanced/metastatic STS has neared completion (NCT02398058), and a phase 2 single-arm trial of olaparib combined with trabectedin in patients with advanced sarcoma has a LMS-specific cohort (NCT04076579). Another phase 2 study is testing the combination of olaparib plus temozolomide specifically in patients with advanced, metastatic, or unresectable uterine LMS (NCT03880019). An overview on selected new treatment strategies is depicted in Figure 1.While asking patients, innovations in therapy and new treatment strategies are certainly of the utmost importance, however, it is necessary to take the whole treatment journey into account. This includes a multimodal strategy and combination of different treatment modalities such as surgery, chemotherapy, radiotherapy or potential new innovative approaches. However, it also encompasses an improved knowledge on treatment sequences depending on the risk of local recurrence and the development of metastasis. Furthermore, there is need for a more personalized approach based on molecular testing. The focus of a treatment strategy should be guided by the patients’ perspective, and a good balance of risks and benefits, and survival gain versus quality of life, respectively.Avoid morcellation: Morcellation of the uterus is a surgical technique which is performed to remove the uterus or leiomyomas through minimally invasive surgical approaches. It may be performed during vaginal, laparoscopic, or abdominal surgery using a scalpel, scissors, or a power morcellator. A commonly used alternative to morcellation of an enlarged uterus is a total abdominal hysterectomy associated with higher morbidity and mortality and diminished quality of life. Morcellation of a malignancy is contraindicated. Although too often not the case, women should be evaluated preoperatively to identify malignancy. Women most often do not undergo a tissue biopsy and diagnosis prior to morcellation; thus, there is a risk that a woman with a presumed leiomyoma may have a malignancy that may be spread through morcellation [44]. Morcellation bears the risk to disseminate tumor cells into the pelvis and peritoneal cavity, with a poorer prognosis as a major consequence [45,46]. The risk of an unexpected LMS diagnosis is estimated to be as frequent as 1 in 498 women. The risks associated with abdominal hysterectomy (blood loss, deep venous thrombosis, death) must be balanced against the risk of unexpected malignancy after morcellation. Existing data support a minimally invasive approach for younger women and procedures that do not involve morcellation for older women. The health care team should engage the patient in shared decision making, including informed consent, explaining the risks and benefits of each approach for presumed leiomyomas, the risks of morcellation, the rationale for a biopsy prior to surgery and alternatives to morcellation [47]. As an example, the sarcoma charity “Sarcoma UK” and the “Royal College of Obstetricians and Gynaecologists” have developed a consent advice and patient information for women offered a myomectomy or hysterectomy using morcellation, in order to enable them to make an informed choice about which surgery is right for them and to encourage discussion of the individual risks of surgery, including the risks of morcellation [48]. The question of many uterine patients with LMS remains whether or not there may be a way to make a definitive or at least more suggestive diagnosis, e.g., through imaging before any kind of invasive measure. Additional research is needed to understand the prevalence of LMS at the time of surgery for presumed leiomyomas, to better define risk factors for LMS, and to develop preoperative diagnostic tools and methods to improve the safety and efficacy of morcellation. Additionally, there is a clear need for more collaboration between gynecologic oncologists and sarcoma experts.Explore the immune system: Monoclonal antibodies that block suppressive functions of immune checkpoint proteins PD1/PD-L1 and CTLA4 have remarkable anti-tumor activity in a subset of patients with STS [49]. Unfortunately, both vascular and uterine LMS respond poorly to checkpoint inhibitors in published clinical trials [50], including anti-PD1/PD-L1 monotherapy [51,52], combined PD1/CTLA4 inhibition [53], or PD1 therapy combined with cyclophosphamide [54] or anti-VEGF TKI axitinib [55], although collectively the numbers of patients with LMS included in these all-comer studies are small. Multiple retrospective studies of immune-related genetic expression have suggested that LMS do have underlying immunogenicity [56,57,58], but the exact therapeutic strategy to exploit this remains elusive. Uterine LMS is being studied in a phase 2 study evaluating nivolumab alone or in combination with ipilimumab in treating patients with advanced uterine LMS from the National Cancer Institute (NCT02428192). Additionally, ongoing clinical trials are combining cytotoxic chemotherapy, including doxorubicin, gemcitabine, and trabectedin with checkpoint blockade, which may help to increase tumor immunogenicity of “cold” tumors. For example, a phase 2 study from the German Interdisciplinary Sarcoma Group (GISG) testing the combined treatment with nivolumab plus trabectedin in patients with metastatic or inoperable STS has a dedicated LMS cohort (GISG-15; NiTraSarc; NCT03590210). Additionally, studies are looking at more effective drugs to repolarize suppressive myeloid cells within tumors, suspected to be a major mechanism of resistance in LMS, including a study of DCC-3014, an anti-CSF1R TKI in combination with avelumab (anti-PD-L1) (NCT04242238). Finally, cabozantinib is being explored in a randomized study with or without dual PD1/CTLA4 checkpoint blockade, with a broader spectrum TKI potentially more impactful to the tumor microenvironment than narrow VEGF inhibitors (NCT04551430). Overall, it is critical to support preclinical and translational laboratory research with these and other ongoing studies to better understand mechanisms of response and resistance in treated patients with LMS, and to develop biomarkers for specific immune subsets of LMS to better tailor combination therapies. Additionally, further transcriptomic work to characterize potential tumor neoantigens and identify responding T cell clones may one day identify novel targets for adoptive cellular strategies.Investigate the role of circulating tumor DNA for matching therapy and as a biomarker of prognosis, response to therapy and minimal residual disease: Circulating tumor DNA (ctDNA) offers a rapid and non-invasive method of next-generation sequencing (NGS) that could be used for diagnosis, prognostic assessment, disease-response assessment to therapy, and detection of recurrence [59,60,61]. In a recent study, 59 of 73 metastatic patients with LMS were found to have >1 cancer-associated genomic alteration. A total of 45 patients were women with a median age of 63 years (range, 38–87). The most common alterations detected were in TP53 (65%), BRAF (13%), CCNE (13%), EGFR (12%), PIK3CA (12%), FGFR1 (10%), RB1 (10%), KIT (8%), and PDGFRA (8%). Additionally, alterations included RAF1, ERBB2, MET, PTEN, TERT, APC, and NOTCH1. Potentially targetable mutations were found in 40% of the 73 patients. A total of 5% were incidentally found to have germline TP53 mutations [62].NGS of ctDNA allows identification of genomic alterations in plasma from patients with LMS [63]. Other than pazopanib with its unknown mechanism of action, there is limited activity of current targeted agents for patients with LMS. These findings underscore the need to develop therapies against TP53, cell cycle, kinase signaling, and epigenetic pathways. Further validation and prospective evaluation is warranted to investigate the clinical utility of ctDNA especially for patients with LMS. A Sarcoma Alliance for Research Through Collaboration (SARC)-funded pilot study is evaluating ctDNA as a biomarker of relapse-free survival and response to therapy in patients with high-grade, high-risk, localized LMS; and a SARC-supported study of ctDNA as biomarker of sarcoma response to chemotherapy in patients with metastatic LMS is currently planned.Implement molecular characterization of LMS-NGS, transcriptome and exome data in order to develop prognostic and predictive markers as well as to design molecularly driven clinical trials: Over the past decade, a significant amount of work has resulted in a new molecular understanding of many sarcoma subtypes including LMS [64,65]. This includes the identification of three molecular subtypes of LMS with distinct transcriptomic profiles and clinicopathological characteristics [42,66,67]. However, unanswered questions remain. We know that LMS of different anatomic sites have different natural histories, prognosis, and responses to therapy, but the molecular characteristics which could differentiate these subtypes remain unknown. NGS of tumor specimens allows identification of specific gene alterations that can aid with tumor classification and suggest potential mutation-specific therapeutic targets or clinical trials. Recently, NGS of 21 LMS from various sites revealed 86 non-synonymous, coding region somatic variants within 151 gene targets (mean of 4.1 variants per case); the most frequently altered genes were TP53 (36%), ATM and ATRX (16%) as well as EGFR and RB1 (12%) [68]. Perhaps a molecular “signature” could serve as a better prognostic and predictive biomarker than the anatomic location. For instance, emerging data from several retrospective studies in LMS have shown that the Complexity INdex in SARComas (CINSARC) and Genomic Grade Index transcriptomic signatures have utility in predicting risk of relapse [69,70,71,72]. CINSARC is currently undergoing prospective evaluation in the peri-operative chemotherapy setting (NCT03805022, NCT02789384 and NCT04307277). Potentially, there could be a molecular marker or gene signature which could help to understand why some patients with LMS respond to ifosfamide while others do not. Moreover, which molecular characteristic could identify super-responders to temozolomide or dacarbazine, or other chemotherapies with occasional exceptional activity in patients with LMS?These questions could be answered using a large-scale genomic and transcriptomic database containing a large number of diverse LMS as well as the corresponding rich clinical data. To date, no such data set exists because current genomic databases have very few patients with LMS and sparse or completely missing clinical data.Essential need for basic research and translational pipeline: Valid laboratory models of LMS are urgently needed to understand LMS-specific oncogenic pathways, facilitate unbiased studies of LMS cancer dependencies, and support translational studies to identify novel therapeutic strategies for this disease. Several reports have identified a critical lack of fidelity to the human disease in epigenetic and transcriptional programs in established LMS cell lines [67,73]. This may arise from a misdiagnosis of the tumor of origin of these cells or the significant heterogeneity within this disease. Alternatively, lack of model fidelity may arise from the characteristic loss of tumor suppressors and absence of recurrent oncogene activation, leading to divergent evolution of LMS-derived cultures over time. Additional efforts at generating, validating, and distributing novel LMS cell lines is essential to future basic research efforts, including unbiased assessments of LMS-specific vulnerabilities arising from CRISPR and chemical dependency screens [74]. Mouse models of LMS have been reported, including genetically engineered mouse models that develop spontaneous LMS-like tumors [75] and LMS patient-derived xenografts (PDX) [76,77]. While there are early reports of the potential value of LMS mouse models in preclinical studies, these and other novel models need to undergo similar scrutiny as cell lines to demonstrate their fidelity to the primary disease. To evaluate new agents that exploit metabolic vulnerabilities or the immune system, consideration of more complex preclinical models (e.g., co-culture systems, syngeneic models and “humanized” mice with immune cell engraftment) would be of value. The ultimate goal of such model development and characterization is to confidently identify and prioritize therapeutic targets to translate into LMS-specific clinical trials.Despite the progress that has been achieved thus far, there remain several outstanding areas that should be the focus of future basic and translational LMS research efforts. With the limited response to current chemotherapy in LMS, there is a need for sustained efforts to define effective targeted therapies in LMS, which may include ATR inhibitors, PARP inhibitors and other DNA damage repair targets, PI3K/mTOR inhibitors [78], metabolic vulnerabilities such as exploiting arginine starvation [79,80] and directed immunotherapy. Furthermore, as omic technologies become more accessible and cost-effective, there should be concerted investigations to determine whether integration of multiomic measurements such as epigenomics, proteomics and metabolomics may yield more robust drug discovery targets and biomarkers [81]. Finally, given the rarity of the disease, there is an urgent need for international collaborations within a coordinated research strategy framework that minimizes overlap and maximizes the limited funding available in the field.Evaluate imaging modalities to better distinguish features of LMS: LMS metastasize with high frequency, and patients with advanced-stage disease have a poor prognosis. LMS may arise in various anatomic sites, but are broadly divided into uterine and extra-uterine tumors. Those found in extra-uterine soft tissues may arise within a vessel, such as the vena cava or renal vein in the retroperitoneum. Absent this association with a major vessel, imaging findings at CT or MRI are non-specific. This is especially problematic within the uterus, where benign leiomyomas may be difficult to distinguish from LMS. Unfortunately, difficulties distinguishing these entities preoperatively may lead to unplanned excisions or morcellation of uterine LMS adversely influencing patient’s outcome (see Chapter 3). It has recently been shown that the presence of at least three qualitative MR imaging features was strongly associated with a LMS: nodular borders, hemorrhage, central necrosis, and “T2 dark” area(s), with other studies emphasizing the importance of nodular contours and central necrosis [82].One advance could be the use of radiomics, which describes the extraction of large amounts of quantitative data from medical images that can be correlated with tumor histology and clinical outcomes. Radiomics entails tumor segmentation using software that subsequently analyzes various image features, yielding first-, second-, and higher-order statistics that describe image signal intensity and spatial heterogeneity. Investigators employing histogram analysis have found that LMS is marked by higher signal intensity voxels (a voxel represents a value, signal intensity in MRI or Hounsfield units in CT, in the three-dimensional image data acquired on MRI and CT scans) on T2-weighted images, specifically the mean of the bottom 10th percentile on histogram analysis [83]. Recent work has suggested that radiomics analysis of the entire uterus, and not just the tumor volume, may yield the best diagnostic performance in discriminating leiomyoma from LMS, and that optimal radiomics models perform comparably to radiologists [84]. These considerations highlight the challenge of characterizing soft-tissue tumors with conventional imaging strategies, and point toward utilization of radiomics and image texture analysis in enabling more complete and accurate uterine tumor characterization.Tumor necrosis as a CT imaging biomarker has recently been shown to improve histologic grading based on core needle biopsy alone, since biopsy may undersample areas of tumor necrosis due to intrinsic tumor heterogeneity and biopsy technique. Furthermore, radiomics features show strong correlation with histologic grade in STS. Development of a composite grading system that accounts for histology, tumor NGS, as well as novel radiomics imaging biomarkers, could lead to improved clinical decision making at the time of initial diagnosis.Because many LMS tend to be biologically aggressive tumors with high metastatic potential and local recurrence rates, systemic therapy plays an important role in the multimodality treatment strategy. The radiologic assessment of the therapeutic efficacy then becomes a crucial task so that ineffective therapies can be switched out for alternative and potentially more active regimens. In STS, it is well known that conventional size-based criteria of disease response may fail to accurately reflect treatment efficacy. This was recently highlighted in a phase 2 study (22/37 with LMS) of sorafenib and dacarbazine that showed RECIST versus Choi response rates of 14% (5/37) versus 51% (19/37), respectively [85]. Such concerns have driven the search for alternate response criteria that incorporate changes in tumor enhancement, and texture analysis of changes in T2 signal intensity may predict neoadjuvant response in STS more accurately than RECIST. Radiomics features are known to be independently associated with OS in STS, even after adjusting for patient age and tumor grade. While computationally abstruse, radiomics features may be particularly well-suited to interrogating treated LMS, where the spatial clustering of enhancing and non-enhancing voxels map histologically to viable and necrotic tumor components. Development of radiologic response criteria that may include radiomics features and are more finely tuned to the biology of STS, and those of LMS in particular, are critical in expediting assessments of drug efficacy. Future clinical trials for patients with LMS should incorporate radiomics in order to better define response to systemic therapy and diagnosis of LMS versus leiomyoma.Improve early detection of diagnosis: Obviously, a timely and correct diagnosis is essential to improve the prospects for survival which in general are not good for patients with LMS and have hardly improved over the past decades. According to the SPAEN research survey, most patients wish for a way to detect LMS earlier and easier. For most patients, the pathway to a correct diagnosis is a long one, with an extreme example of five years. However, early diagnosis can save lives: approximately 40% of STS are diagnosed in a locally advanced or metastatic stage [86]. While the 5 year survival rate is 81%, if the disease is caught in an early stage, it drops dramatically to 16% if diagnosed in an advanced metastatic stage. Consequently, there is a high need for (a) improved information and education for general practitioners to recognize sarcomas (symptoms and triggers for sarcomas including LMS) and transfer the patient to a specialized center in case of suspect for sarcoma, as well as (b) improved diagnostic measures to confirm the sarcoma diagnosis more easily and faster. This could be achieved through improved imaging, education and training of radiologists and new, innovative ways to detect LMS such as blood tests/ctDNA.Identify patient-reported outcomes (PROs): There is growing recognition in medical oncology of the potential value offered by patient-reported outcomes (PROs) partly fostered by the growth of patient involvement in research and in the provision of health care services. In a review in 1989, Maguire and Selby [87] noted an ambition: “*A multidimensional scale which is specific to patients with cancer meets all the assessment criteria and provides scores which have relevance to clinical judgement remains to be developed”.* Thirty-one years later, we can say that this has now been achieved. However, different tools co-exist which do not allow comparison, confusing and discouraging patients. A multidimensional scale which is specific to sarcoma remains to be developed, although that work is underway. There is some distance still to go to have one which is specific to LMS.Validated composite tools to gather the multidimensional data which enable a health-related quality of life (HRQoL) to be assessed are available, but their main weakness is that they measure a ‘moment in time’ rather than give a full picture of patient experience. The world of ‘quality-of-life’ in research has begun to shift. A fuller understanding of the domains within quality-of-life has grown and it is now possible to construct questionnaires exploring greater detail of specific aspects of the patient experience. This has opened up the importance of individual PROs. Item libraries are now available; typically, the EORTC Quality of Life Group Item Library [88] contains over 900 PRO items, each of them in many languages and all validated. A valuable development has been the PRO Common Terminology Criteria for Adverse Events (CTCAE) from NCI [89]. The CTCAE has been a mainstay of cancer clinical trial practice and reporting for many years, but the grading relied on clinician observation of patients’ experience. The PRO version calls for patients to report their experience first-hand. Gathering these data using smartphones and internet reporting opens the way for a more sensitive and often more accurate reporting of adverse events. Importantly, systems can be set up to offer ‘red-flag’ reports to clinicians, enabling a prompt intervention, sometimes even before the patient is aware a problem is developing. How do we grasp all this development and use it to create a better understanding of the needs of patients with LMS and the opportunities for treating them better? A patient might express this question somewhat differently. What have the doctors been missing about my experience as a patient and how important is it? The absence of a specific sarcoma tool is important and we should support the development work which is underway by NKI Amsterdam together with EORTC and that of UCLH in London. The developers are aware of specific aspects of LMS patient demographics (uterine, limb, RPS, etc.) and are allowing for these specific presentations.What we can do in the interim is take advantage of the existing work with the PRO CTCAE [90,91] and explore its use in our research. The development of patient input systems using smartphones, tablets and internet links would also open the way for longitudinal assessment where appropriate, allowing trends to be illustrated rather than single scores at absolute ‘moments in time’. We can also identify and share experience with use of individual PROs from Item Libraries such as EORTC. Identifying specific PROs which add value to our understanding of the LMS patient experience would be a valuable step forward.The need to identify PROs which offer value in the LMS setting calls for patients to be involved in the discussion. It would support moves to use PROs if researchers established a consultative patient group to work with them on research design issues, not necessarily exclusive to PROs, but with a focus on helping determine the experience issues which matter to patients. It can be argued that seeking PROs without taking account of patient inputs would be unethical. The patient viewpoint is that the longer-term aim should be to develop synchronous clinical and patient-reported pathway information which can inform clinicians and patients from diagnosis, through treatment, witnessing the options available and the choices which can be made at any point on that pathway [92]. Such a pathway would also have value to regulators and health care funders. In a rare disease such as LMS a well-attested pathway may also offer comparisons which can be difficult to achieve when numbers limit the practicality of traditional randomization.Communication, information and support: The disease is rare and therefore unknown to most people. Reason enough for patients with LMS to feel lonely and have a great need for information about their disease and possible treatments as well as support. This need for information and support is a holistic one for patients, as this disease is not just about treatment and therapy options but a complete change in lifestyle and priorities. Thus, patients would like to see a clear case management during treatment, which pays attention to the patient as a human being, opposed to just the tumor. This also includes psycho-social support, better physio and recovery care. Additionally, this need is not only relevant at a single point of time, but it is a long-term, ongoing and changing need during the entire patient journey. This starts with better information at the time of diagnosis, including the possibility to ask for a second opinion, information on sarcoma specialists and expert centers for sarcomas, as well as information on how the disease and the treatments will impact the life of the patient (work, family and partnership) and how to cope with the burden of this disease. Furthermore, many patients ask for information on what they can do themselves to improve their quality of life and to strengthen their physical as well as mental well-being. This involves questions or changes in lifestyle, diet or exercise. While patient advocacy groups offer a wide range of practical information, the treating doctor or treatment team is considered to be a partner and an important source for information and is therefore expected to be able to support the patient. Joint decision making during treatment is important but requires adequate information on the different options for treatment. In this respect patients also mention that this should look at the entire trajectory of treatment, not just the next step. It also requires an open and honest communication between both parties.Address survivorship and end of life: Those two topics seem to be contrary, but they share challenges. Both might be considered to be the “forgotten” parts in sarcoma management. While sarcoma survivors must cope with having been a patient and returning to a “new normal”, they often feel left alone and insecure. Not only is the distinction between “patient” and “survivor” or “healed person” unclear. Follow-up procedures (if any) are very often not specified, in terms of frequency, intervals and extensiveness. Furthermore, dealing with the potential risk of late effects is very often burdened on the patient. “End of life”, on the other hand, is a delicate, but important question for many patients. The thought of death is scary, but for a lot of sarcoma patients it is unfortunately part of their story. Questions on how the last part of the journey will look like, if they have choice of where to die and how to tell their families are important topics and should and need to be addressed in a sensitive way.

## 8. Conclusions

In summary, individualized biology driven treatment must be the standard of care in malignancies such as LMS. It is strongly advisable to seek therapeutic advice of a high-volume reference center or to enroll patients with LMS in suitable subtype-specific clinical studies, factors clearly linked to a more beneficial outcome for this patient group [93,94,95]. The following summarized aspects depicted in Figure 2 may help to drive forward beneficial treatment strategies for patients with LMS including the most important—the patient’s voice:

## Figures and Tables

**Figure 1 cancers-13-00886-f001:**
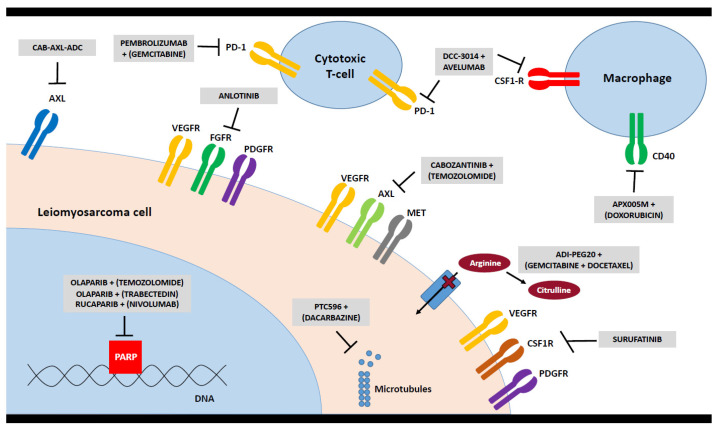
New possible treatment strategies and their mode of action for patients with LMS.

**Figure 2 cancers-13-00886-f002:**
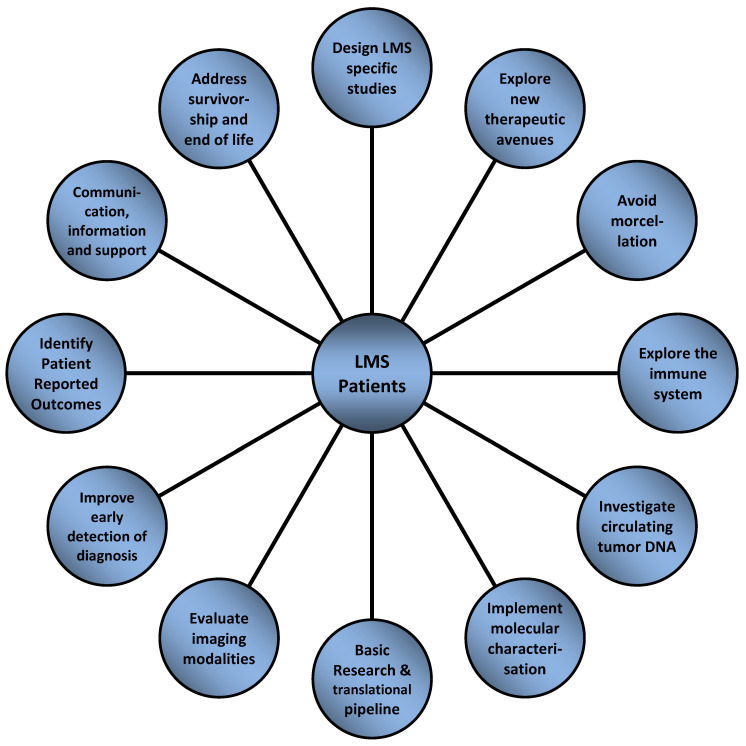
Future perspectives for the management of patients diagnosed with LMS.

**Table 1 cancers-13-00886-t001:** Overview of key studies on current clinical management of advanced/metastatic uterine and extra-uterine patients with leiomyosarcomas (LMS) in the context of soft-tissue sarcoma (STS) therapy.

Agent(s)	Phase	*n*	Line	ORR		PFS (Months)		OS (Months)	
**All STS**									
Doxorubicin vs. Doxorubicin + Ifosfamide [15]	III	455	1st	14%	26%	4.6	7.4	12.8	14.3
Doxorubicin vs. Gemcitabine + Docetaxel [28]	III	257	1st	19%	20%	5.4	5.5	17.6	15.5
Gemcitabine vs. Gemcitabine + Docetaxel [25]	II	122	1st–3rd	8%	16%	3.0	6.2	11.5	17.9
Dacarbazine vs. Gemcitabine + Dacarbazine [29]	II	113	2nd+	25% ^a^	49% ^a^	2	4.2	8.2	16.8
Pazopanib vs. Placebo [31]	III	372	2nd+	6%	0%	4.6	1.6	12.5	10.7
**LMS**									
Doxorubicin + Dacarbazine [16]	retro	22	1st			15.1		33.9	
Gemcitabine + Docetaxel [39]	II	45	1st	25%		7.1		17.9	
Trabectedin vs.Dacarbazine [40]	III	403	3rd+	10%	7%	4.8	1.5	14.1	13.6
**uLMS**									
Gemcitabine + Docetaxel [35]	II	42	1st	36%		4.4		16.1	
Gemcitabine + Docetaxel [34]	II	51	2nd+	27%		6.7		14.7	
Trabectedin vs. Dacarbazine [37] ^b^	III	232	3rd+	11%	9%	4.0	1.5	13.4	12.9
Pazopanib vs. placebo [38] ^b^	III	44	2nd+	11%	0%	2.9	0.8	17.5	7.9

Abbreviations: ORR, overall response rate; PFS, progression-free survival; OS, overall survival; ^a^ clinical benefit rate including stable diseases; ^b^ subset analysis of a randomized trial.

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
