# Peer review of "Unmet Medical Needs and Future Perspectives for Leiomyosarcoma Patients—A Position Paper from the National LeioMyoSarcoma Foundation (NLMSF) and Sarcoma Patients EuroNet (SPAEN)"

_cancers, 2021, doi:10.3390/cancers13040886_

Round 1

Reviewer 1 Report

Very comprehensive review of the state-of-the art treatment and of research perspectives for leiomyosarcoma (LMS).

I would like to see what the authors have to say about improved local control in LMS from a research standpoint: newer surgical or radiation techniques, for instance.

Reviewer 2 Report

Overall this is complete, thorough, and unbiased. 

The authors have done an excellent job in summarizing the available data, as well as describing the future perspectives and unmet medical needs in this patient population.

I have a few comments:

Please use respectful use of language towards patients, instead of leiomyosarcoma patients, please use patients WITH leiomyosarcoma or people with leiomyosarcoma.

A small paragraph on special populations, such pre-menopausal women, might be useful.

More details about the biology of the disease (pathology, molecular diagnostics, natural disease history) might better help orient the reader.

Is the figure 1 an original figure, or adapted?

Reviewer 3 Report

This is a updated position paper about leiomyosarcoma. Although the paper is already very extensive, information regarding leiomyosarcoma predisposition genetic syndromes such as Li Fraumeni and Hereditary leiomyomatosis (HLRCC) syndromes are lacking and should be discussed. 

Specific points:

line 43: LMS typically occur in middle aged or older adults (consider comment about its appearance in young patients, specially in Li Fraumeni patients)

line 52: LMS are malignant mesenchymal tumors that originate from smooth muscle (this concept of origin is very controversial, so its better to avoid this affirmation; consider put "LMS are smooth muscle malignant mesenchymal tumors 

line 71: explain what R1 and R2 resection mean

line 73: The 5-year local and distant recurrence rates fo primary LMS are 10-20% and 30-40%, respectively. ( complete with: for high grade tumors)

line 120: "The use of preoperative diagnostic biopsy is rarely utilized although could impact treatment recommendations..." (The authors should rephrase this sentence to not give the impression that a biopsy should be performed in case of leiomyosarcoma suspicious. Biopsy for uterine leiomyosarcoma is not recommended due to the lack of sensibility in detecting malignancy)
